# The Use and Utility of Machine Learning in Achieving Precision Medicine in Systemic Sclerosis: A Narrative Review

**DOI:** 10.3390/jpm12081198

**Published:** 2022-07-23

**Authors:** Francesco Bonomi, Silvia Peretti, Gemma Lepri, Vincenzo Venerito, Edda Russo, Cosimo Bruni, Florenzo Iannone, Sabina Tangaro, Amedeo Amedei, Serena Guiducci, Marco Matucci Cerinic, Silvia Bellando Randone

**Affiliations:** 1Department of Clinical and Experimental Medicine, University of Florence, 50134 Florence, Italy; francesco.bonomi@unifi.it (F.B.); silvia.peretti@unifi.it (S.P.); gemma.lepri@unifi.it (G.L.); edda.russo@unifi.it (E.R.); cosimo.bruni@unifi.it (C.B.); amedeo.amedei@unifi.it (A.A.); serena.guiducci@unifi.it (S.G.); marco.matuccicerinic@unifi.it (M.M.C.); 2Rheumatology Unit, Department of Emergency and Organ Transplantations, University of Bari Aldo Moro, 70121 Bari, Italy; vincenzo.venerito@uniba.it (V.V.); florenzo.iannone@uniba.it (F.I.); 3Department of Rheumatology, University Hospital of Zurich, University of Zurich, 8006 Zurich, Switzerland; 4Department of Soil, Plant and Food Sciences, University of Bari Aldo Moro, Istituto Nazionale di Fisica Nucleare, Sezione di Bari, 70121 Bari, Italy; sabina.tangaro@uniba.it; 5Unit of Immunology, Rheumatology, Allergy and Rare Diseases (UnIRAR), IRCCS San Raffaele Hospital, 20132 Milan, Italy

**Keywords:** systemic sclerosis, machine learning, artificial intelligence, precision medicine

## Abstract

Background: Systemic sclerosis (SSc) is a rare connective tissue disease that can affect different organs and has extremely heterogenous presentations. This complexity makes it difficult to perform an early diagnosis and a subsequent subclassification of the disease. This hinders a personalized approach in clinical practice. In this context, machine learning (ML), a branch of artificial intelligence (AI), is able to recognize relationships in data and predict outcomes. Methods: Here, we performed a narrative review concerning the application of ML in SSc to define the state of art and evaluate its role in a precision medicine context. Results: Currently, ML has been used to stratify SSc patients and identify those at high risk of severe complications. Additionally, ML may be useful in the early detection of organ involvement. Furthermore, ML might have a role in target therapy approach and in predicting drug response. Conclusion: Available evidence about the utility of ML in SSc is sparse but promising. Future improvements in this field could result in a big step toward precision medicine. Further research is needed to define ML application in clinical practice.

## 1. Introduction

Systemic sclerosis (SSc) is a rare connective tissue disease characterized by autoimmune features, vasculopathy, and fibrosis [1]. Among rheumatic diseases, SSc has the highest mortality rate, due to the fact that treatment options do not address both the fibrotic and inflammatory disease features [2,3]. Skin fibrosis is a well-known hallmark of the disease and the extension of skin involvement has been proven to influence disease-associated mortality [4]. However, many different organs can be involved in the course of the disease, including heart, lungs, kidneys, and gastrointestinal tract. Pulmonary complications are the most common cause of death in SSc, with interstitial lung disease and pulmonary arterial hypertension being the most common manifestations, occurring in approximately one-third of cases and being associated with reduced survival [2].

Since SSc is an uncommon disease with multiple heterogeneous symptoms, early diagnosis and predicting the risk of disease progression pose a significant challenge for physicians [5,6]. In fact, SSc is characterized by a wide variability in both the clinical phenotype, the genetic expression, the autoantibody pattern, and disease evolution, limiting the progress in research due to relevant patient heterogeneity and imperfect outcome measurements [7]. The disease is still classified according to skin fibrosis extension in a limited and a diffuse cutaneous subset, although recently it has become clearly evident that this subclassifying method may not be able, in clinical practice, to predict the risk of progression [8,9]. For this reason, we searched for new methods able to combine multiple variables with the final aim to provide a patients’ outcome and risk stratification. Promising help could come from the use of artificial intelligence and specifically the machine learning model.

Machine learning (ML) is a branch of artificial intelligence (AI) that uses algorithms to recognize relationships in data and its use in the medical field is increasing [10]. An ML approach includes supervised and unsupervised model learning [11]: the former is constructed to predict known groups or values, while the latter is used to find associations and patterns out of raw data that results in groups of similar samples. As a general rule, ML algorithms try to discover underlying and unanticipated connections in data, helping to generate hypotheses [11].

Until now, ML has been used in various areas of bio-medicine for different purposes such as the analysis of medical images [12], the subclassification of patient cohorts [13], the prediction of drug response [14], and to guide personalized medicine [15]. Of interest, ML has also been used in gut microbiome research and has been useful to predict atherosclerotic cardiovascular disease [16] and to predict chemotherapy effectiveness and tolerability in the oncology setting [17]. In rheumatology, ML approaches have been studied so far to aid in the identification of new possible therapies in primary Sjogren’s syndrome and systemic lupus erythematosus (SLE) [18], and to classify patients or predict disease outcomes based on genetic data [19]. When deployed on data from patients with inflammatory arthritis, supervised ML algorithms have proven effective in predicting treatment responses or for the automatic histopathological grading of synovitis [20,21].

To date, few studies have reported the use of ML in SSc as a possibility to overcome many current limitations in diagnosis and treatment. It is conceivable that SSc might need a multidimensional approach to build predictive models. The superiority of ML methods considering nonlinear relationships is crucial as it could refine our *modus cogitandi* when dealing with the complexity of SSc [22]. In fact, it could help physicians predict organ involvement and allow a customized treatment according to genetics, autoantibody profile, and organ involvement, and help in finding new biomarkers to tailor the follow-up [23]. The aim of this paper was to summarize the current evidence on the use of ML in SSc and evaluate its possible role in precision medicine through a narrative review of the literature.

## 2. Materials and Methods

We performed a nonsystematic (PRISMA protocol not followed) narrative literature review on PubMed with the following MeSH terms: [scleroderma, systemic AND machine learning]; [scleroderma, systemic AND machine intelligence]; [scleroderma, systemic AND artificial intelligence]; [scleroderma, systemic AND deep learning]. No time limit was put on the research and each paper’s reference list was checked for additional studies.

## 3. Results

Our literature search retrieved articles published between 2015 and 2022. The data derived showed that ML has found possible applications in different fields, ranging from the subclassification of SSc patients to predictions of treatment response. A summary of the analyzed works is reported in Table 1.

### 3.1. ML in the Stratification of SSc Patients

The natural course of disease is extremely heterogeneous and the management of SSc patients remains a great challenge. Hence, the need for clinicians to stratify patients according to the risk of disease progression and organ involvement. The common goal of the scientific community is to develop a precision medicine model with a tailor-made follow-up able to identify progressors early and limit over-testing. Currently, there are no evidence-based guidelines regarding the intensity and the frequency of follow-up in SSc patients [23], except for the management of pulmonary arterial hypertension (PAH) [36].

In this framework, ML may integrate several data and it may be helpful for clinicians to identify high-risk patients, with few but promising studies available so far.

Van Leeuwen and colleagues [24] successfully created a prediction model able to identify nonprogressors. First, they defined progression as worsening in one or more organ systems and/or start of immunosuppressive therapy or death between two visits. Patients with none of these events were labeled as ‘non progressors’. Consequently, they considered a group of patients with at least three assessments and with a complete organ evaluation available (such as: modified Rodnan Skin Score (mRSS), blood tests, 24 h ECG recording, echocardiography, pulmonary function test, high-resolution chest tomography, cardiopulmonary exercise test). A total of 90 variables derived from the assessments were approached through a machine learning model able to extract signs of progression. Patients were classified as high, intermediate, and low risk of progression and they were compared with real outcomes after a period of 5.4 years. The results of this study showed that the model perfectly recognized patients at low risk of progression with a correctness of 100%; in fact, none of the low-risk patients really experienced worsening of disease. By contrast, less than half of the patients identified by ML as being high-risk really progressed. These results may have implications in clinical practice, as low-risk patients might be assessed with less intensive follow-up protocols, therefore reducing the burden of health care cost with minimal risks of missing organ progressors.

Other authors proposed a classification based on skin transcriptomic data, with the rationale that a deeper understanding of SSc pathogenesis could better define patient subgroups. The aim was to identify specific molecular signatures that can be related to the disease course. In this framework, ML was used to integrate genetic and clinical data with molecular phenotype with the final goal to predict disease outcome. Multiple studies have already identified four molecular subsets through gene expression in skin biopsies [37,38,39,40]: normal-like, limited, fibroproliferative, and inflammatory. These subsets combine a plethora of cytokines (such as IL-5, IL-6, IL-13, IL-18, PDGF, etc.) playing a role in SSc pathogenesis [41,42]. Of note, Frank et al. [25] identified a classifier unsupervised machine learning algorithm able to assign a single skin sample to a specific gene expression subset. Their study included 297 microarrays from 102 patients, presenting genes derived from all the expression subsets. The four subsets were all homogeneously represented, except for the limited subtype that was underrepresented. They found 245 genes expressed in the inflammatory subset, mostly related to immune system response, and 245 genes in the fibroproliferative one with high specificity (95.8% vs. 94.1%, respectively) and good sensitivity (83.3% vs. 89.7%, respectively). In contrast, the limited subset showed lower sensitivity, likely due to the limited size of the sample. This study is proof that a set of genes might identify a specific patient subset, with potential implications in prognosis and therapy. For example, the inflammatory subset is characterized by the activation of the immune system and seems to be associated with a better response to disease-modifying antirheumatic drugs (DMARDs) [38].

Xu and colleagues [26] recently identified 80 pathway signatures that could stratify patients into eight subtypes. They analyzed microarrays from 221 involved skin samples of 141 SSc patients at the time of diagnosis and 80 healthy controls. An ML-assisted model identified 80 pathways that could be summarized in five mean pathways: Metabolism-1, Metabolism-2, Immune-fibrosis, Immune Response-1, and Immune Response-2. These five pathway modules could differentiate the eight clusters of disease, built through different combinations of their gene expressions. Clusters 1 and 6 were characterized by metabolic pathways (Metabolism-1 and Metabolism-2) and they were similar to the control cohort, although cluster 1 showed more immune activation than cluster 6 and control cohort. Clusters 2 and 4 were characterized by inflammation and immune response activation with high levels of IL-17 and IL-22. Cluster 3 was characterized by an immune-fibrosis pathway. Clusters 5 and 8 exhibited a mixture of inflammatory, pro-fibrotic, and metabolism signatures, and lastly cluster 7 was identified as a separate normal-like cohort with low expression of Metabolism-1 and high expression of Metabolism-2. From the different clusters, the authors managed to underline different cells involved in pathogenesis, with clusters 1 and 2 enriched with myeloid cells and macrophages, cluster 3 enriched with fibroblast and endothelial cells, and cluster 4 with strong expression in all four types of cells. This study highlighted different molecular signatures and pathways, leading the way to a tailor-made treatment and a personalized assessment.

Another study [27] analyzed 48 forearm skin samples from 26 SSc patients treated with nilotinib or with belimumab, integrating molecular data with clinical data such as mRSS, blood tests, FVC, and patient-reported outcomes. The aim was to define a molecular signature at treatment initiation and 52 weeks after, in order to identify improving and nonimproving patients using an ML approach. Samples were evaluated by considering seven histologic features, among which CD34+ and aSMA appeared to be the most predictive elements. This study shows that samples with high aSMA and low CD34+ had higher inflammatory gene expression and higher mRSS. In contrast, low aSMA and high CD34+ were compatible with the normal-like subset. The machine learning model identified CD34+ as a predictor of fibroproliferative and normal-like subsets, whereas aSMA was predictive of the inflammatory subset.

### 3.2. ML Algorithms to Diagnose and Evaluate Lung Involvement

The assessment of lung involvement in SSc currently includes pulmonary function tests (PFTs) and high-resolution computerized tomography (HRCT). Although the latter is the gold standard for ILD detection, there are no clear recommendations about its use and timing to repeat it [43]. Nowadays, clinicians perform HRCT when PFTs show a decline or respiratory symptoms worsen, with the result that some patients might be missed and some others over-tested [44]. In this context, ML algorithms might detect pulmonary involvement before any functional sign of deterioration occurs, with the benefit of improving survival, reducing health care costs, and relieving an unnecessary burden for patients, including radiation exposure.

Murdaca et al. [28] developed a risk-free screening test using ML algorithms, with the aim to predict early pulmonary involvement in asymptomatic patients. They collected clinical data including PFT from SSc patients: seven parameters were identified as the most performant in predicting the presence of lung involvement. Total Lung Capacity (TLC) was estimated to be the best marker, mostly if taken along with forced expiratory volume in the 1st second (FEV1), forced vital capacity (FVC), diffusion lung carbon monoxide (DLCO), and impedance pH monitoring.

Andrade et al. [29] studied the role of the Forced Oscillation Technique (FOT) to detect resistance and reactance in respiratory dynamics. This technique is considered complementary to spirometry [45] because it evaluates different respiratory parameters, with the advantage that it requires minimal cooperation since forced expiratory maneuvers are not needed. The authors proposed an ML approach able to examine lung function data coming from respiratory oscillometry tests.

The extension of lung involvement on CT images has been acknowledged as an independent predictor of mortality [46,47] regardless of the radiological pattern. Chassagnon and colleagues [30] conceived an automated deep-learning-based model to quantify lung involvement, overcoming the variability of a visual approach [48,49]. A machine learning algorithm was trained with reported CT images to determine the extension of the ILD and subsequently a group of patients, with at least two CT of the chest and PFT tests, were included in the analysis. They then compared assessments of ILD extension coming from ML with the ones reported by three qualified radiologists. This algorithm proved to be an accurate and reproducible tool able to quantify lung extension with great accuracy and showing a good correlation between lung involvement and PFT values, surely higher than the one reported for visual assessment.

ML is also rapidly emerging as a powerful tool for radiomic analysis. Radiomic analysis represents a method for the quantitative description of medical images able to describe the tissue in terms of its intensity, texture, and advanced statistical properties through computationally retrieved quantitative data derived from medical images. The added value of radiomic analysis lies in the ability to capture tissue phenotypes on different spatial scales ranging from the radiological/macroscopic to the molecular/microscopic levels [50]. Schniering et al. [35] used an ML approach for radiomic analysis of lung CT in SSc patients to identify homogeneous imaging-based ILD clusters. The analysis produced two distinct and stable patient clusters based on their radiomic profiles. The differences in clinical characteristics were substantial, with patients in cluster 2 (*n* = 31) having a significantly more impaired lung performance in the 6 min walk test and a higher frequency of pulmonary hypertension than patients in cluster 1 (*n* = 59). Cluster 2 was also significantly enriched for honeycombing as a radiological sign of more severe fibrotic lung remodeling [35].

### 3.3. Early Detection of PAH with ML

PAH is a serious complication of SSc which can occur in 8–12% of patients, leading to death in 25–40% of cases [51,52,53]. The gold-standard diagnostic tool for this complication is right heart catheterization (RHC) [54]. The early diagnosis and treatment of milder forms of PAH can lead to better prognosis and increase survival. Nowadays, several tools are available for screening of PAH such as echocardiography, PFT with DLCO, and blood biomarkers (N-terminal prohormone of brain-natriuretic peptide -NT pro-BNP, uric acid). The lack of specificity or sensitivity of the isolated evaluations is partially overcome through their combination into multidomain algorithms [55].

Many authors have shown interest in finding tools able to identify patients with high risk of PAH early, in the framework of precision medicine. Bauer et al. [56] proposed a proteomic screening panel that may incorporate the DETECT algorithm, a two-step noninvasive prediction score, with the potential to enhance specificity. They collected clinical data and serum samples from the DETECT cohort and identified a panel of eight proteins whose expression could discriminate PAH patients from non-PAH patients with the aid of ML. The identified proteins have different roles in the context of PAH: some of them play a role in pulmonary vascular remodeling (RAGE, MMP-2); others are indicative of cardiac dysfunction (NT pro-BNP and IGFBP-7), whereas others are involved in angiogenesis (Collagen IV, endostatin, IGFBP-2, neuropilin-1). Interestingly, this protein panel showed higher accuracy in identifying PAH-SSc patients than NT pro-BNP alone, suggesting that the ML algorithm may detect PAH before the onset of cardiac stress.

### 3.4. ML and Tailored SSc Treatment

The treatment of SSc is still a challenge since most currently available drugs target a single component of the disease pathogenesis (either the vasculopathy, the autoimmune, or fibrotic features). [57] For this reason, every SSc patient should receive a personalized targeted regimen according to the organs involved, the prevalent pathogenic pattern, and the genetic expression. ML can be a very promising tool in these cases because it could help in predicting the response to different drugs. At the moment, few studies have reported the use of an ML approach to personalize SSc treatment.

One study used an ML approach in a meta-analysis [31] to analyze the gene expression data from skin biopsies of patients with SSc before and after five different treatments (Mycophenolate Mofetil (MMF), Rituximab, abatacept, nilotinib and fresolimumab) and to evaluate the treatment response in the skin according to the variations in mRSS over time. Gene expression analysis before and after treatment showed that all therapies except for fresolimumab (an anti-TGFβ monoclonal antibody) modulated immune response-related hallmarks (i.e., IL6/JAK/STAT3 signaling and TNFA/NFKB signaling). The ML algorithm showed that patients who did not respond to fresolilumab had an increased expression of inflammatory genes in the skin biopsy at baseline, suggesting a potential benefit from an immunosuppressant such as MMF. MMF responders, in fact, showed low levels of these inflammatory genes (genes involved in lymphocyte aggregation and type I interferon production). This approach allowed researchers to understand whether patients who failed one therapy could benefit from a different one.

Ebata et al. [32] used ML to analyze the skin sclerosis response in a cohort of 54 patients who received either rituximab or placebo. The study was a double-blind, parallel-group comparison in patients with SSc and showed the efficacy of rituximab on skin sclerosis. An ML approach was used to analyze twenty-seven baseline factors (including sex, different SSc-related signs and symptoms, previous treatments, autoantibody patterns, peripheral CD19-positive cell counts, blood test parameters, and lung involvement) to identify subpopulations with different degree of response to rituximab based on mRSS change at 24 weeks. The algorithm showed that SSc patients with high CD19-positive cell counts (≥57/μL) and high mRSS (≥17) are expected to have greater improvement in mRSS with rituximab. Another interesting subgroup that showed good treatment outcomes was characterized by high CD19-positive cell counts, low mRSS (<17) and serum Surfactant Protein-D (SP-D) levels ≥151 ng/mL; this final variable may hence help in predicting the treatment effect.

An ML approach was also used to predict rituximab responsiveness in SSc-related PAH (SSc-PAH) [33]. Data from 57 SSc patients with PAH were collected in a multicenter, double-blinded, placebo-controlled study which randomized participants to receive two infusions 14 days apart of 1000 mg rituximab or a placebo. The ML algorithm showed that low levels of rheumatoid factors IL-12 and IL-17 were sensitive and specific predictors of a better rituximab response, as measured by an improved 6-min walking distance.

ML also predicted responses to hematopoietic stem-cell transplants (HSCT) in 63 severe SSc cases by using data on gene expression from peripheral blood cells (PBCs) [34]. Patients were divided into two groups: one was treated with cyclophosphamide (CYC) while the other received HSCT. PBCs were collected at baseline and at follow-up at different times after transplant (8, 14, 20/26, 38, 44/48/54 months). Participants who completed any treatment protocol were stratified by intrinsic gene-expression subsets at baseline, evaluated for event-free survival (EFS), and analyzed for differentially expressed genes (DEGs). The ML algorithm showed that patients from the fibroproliferative subset receiving HSCT experienced a significant improvement in EFS with respect to the ones who were treated with CYC (*p* = 0.0091). On the other hand, patients from the normal-like subset or the inflammatory subset did not show a significant difference in EFS regardless of treatment allocation. ML was therefore helpful in determining that just the fibroproliferative subset shows a long-term benefit from HSCT compared to CYC.

## 4. Discussion and Future Perspectives

This narrative review on the use of ML in SSc highlighted many advantages we could obtain from the use of AI in either the diagnostic or the therapeutic approach to the disease (Figure 1). Predicting the future of ML is difficult, as the time scale at which developments will occur is very hard to estimate [58].

To date, the main use of ML has been in oncology-targeted therapy to predict response to chemotherapies and personalize the treatment [14]. In rheumatology, ML has been mainly utilized in nonrare diseases, such as rheumatoid arthritis and SLE, to diagnose patients early at disease onset, identify the ones at higher risk of developing internal organ complications, and find the most effective personalized treatment. The application of ML in this field could provide useful information to guide treatment and predict drug response; however, at the moment, this promising field is largely unexplored [59]. The use of ML in SSc borrowed many concepts and applications from the abovementioned conditions and its future probably rests in their findings, as ML needs a big amount of data to find correlations and connections, making its application in rare diseases slower.

ML could become an extremely helpful tool in personalizing the follow-up of SSc patients, as it could quantify the individual risk of developing specific organ complications and subclassify patients according to their risk of disease progression or the target involved organ. In this regard, an ML approach is a potential game-changer, given its capability of handling both linear and complex nonlinear relationships between patient attributes that are hard to model with traditional statistical methods, especially in complex diseases such as SSc. These modern approaches might allow physicians to both personalize the follow-up (as to which time interval to monitor tests and which exams to make according to the organ more at risk of involvement) and to target the therapy according to both the predicted clinical course and the drug to which each patient could respond (depending on genes expressed at biopsy, blood biomarkers, and instrumental examinations). ML provides promising tools for analyzing large amounts of data (clinical characteristics, laboratory results, treatments, and outcomes) and could allow us to discover new markers to determine the disease outcomes or drug responses. Additionally, ML use in comparative imaging could allow physicians to detect abnormalities that would otherwise be missed and could help to differentiate relevant from nonrelevant changes.

However, barriers to the adoption of ML in rheumatology do exist. One of the reasons emphasized in much of the literature is the role of complementary innovations in the successful adoption of ML and other information technology by companies. The performance of ML algorithms is also contingent on the quality of the data available. Thus, a second barrier to adoption is limited access to data. Medical data are often difficult to collect and difficult to access. Rheumatologists often resent the data-collection process when it interrupts their workflow, and the collected data are often incomplete [60].

Finally, another barrier is that ML and AI models are selected with the criterion of maximizing accuracy. This results in the selection of more complex models and often the loss of transparency of predictive outputs.

The present review’s main limitation is in the nature of the method: being a narrative review, it is too subjective and lacks a systemic approach.

## 5. Conclusions

AI and ML have been used successfully in different medical fields, from oncology to rheumatology. Despite its application in SSc being still sparse, the data collected up until now are promising. Applications of ML are potentially going to increase with time, proving its usefulness in supporting physicians towards personalized medicine.

## Figures and Tables

**Figure 1 jpm-12-01198-f001:**
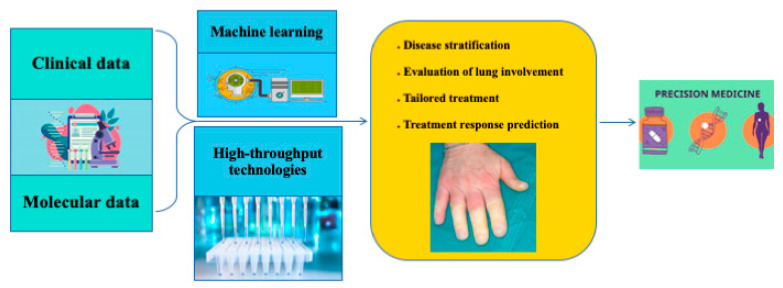
Overview of ML application in SSc to help in precision medicine.

**Table 1 jpm-12-01198-t001:** Summary of analyzed works regarding ML use in SSc.

Authors	Year of Publishing	Journal	No. of Patients	Aim of ML Use
**Leeuwen N.M. van, et al.** [24]	2021	*RMD Open*	248	To predict risk of disease progression in order to develop a tailor-made follow-up
**Franks J.M., et al.** [25]	2019	*Arthritis Rheumatol.*	102	To identify specific molecular signatures from skin biopsies which can be related to disease outcome
**Xu X., et al.** [26]	2020	*PLoS ONE*	221	To identify molecular pathways from skin biopsies in order to obtain a finer SSc stratification
**Showalter K., et al.** [27]	2021	*Ann. Rheum. Dis.*	26	To identify molecular signatures able to predict the treatment response (improvers vs. nonimprovers)
**Murdaca G., et al.** [28]	2021	*Diagnostics*	38	To predict early pulmonary involvement in asymptomatic patients
**Andrade D.S.M., et al.** [29]	2021	*Biomed. Eng. OnLine*	82	To examinate lung function data coming from respiratory oscillometry test
**Chassagnon G., et al.** [30]	2020	*Radiol. Artif. Intell.*	208	To quantify lung disease extension from HRCT images
**Taroni J.N., et al.** [31]	2017	*J. Invest. Dermatol.*	Meta-analysis (total 35)	To evaluate gene expressions on skin biopsies and predict response to different treatments
**Ebata S., et al.** [32]	2022	*Rheumatol. Oxf. Engl*.	54	To find possible predictors of favorable response to RTX
**Zamanian R.T., et al.** [33]	2021	*Am. J. Respir. Crit. Care Med.*	57	To evaluate RTX response in SSc-related PAH
**Franks J.M., et al.** [34]	2020	*Ann. Rheum. Dis.*	63	To evaluate stem cell response in severe SSc
**Schniering J., et al.** [35]	2022	*Eur. Respir. J.*	118	To identify homogeneous imaging-based ILD clusters through a radiomic analysis of lung CT in SSc patients

## Data Availability

Not applicable.

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
