# Peer review of "The Use and Utility of Machine Learning in Achieving Precision Medicine in Systemic Sclerosis: A Narrative Review"

_jpm, 2022, doi:10.3390/jpm12081198_

Round 1
Reviewer 1 Report
Bonomi et al. described through a narrative review the state of the art of machine learning approach in systemic sclerosis. The review is clearly presented and it is informative on a interesting topic in the field.
As a minor request I suggest a sentence in the discussion session about limitations of ML. For example the importance of quality of data to obtain reliable results.
Author Response
Dear reviewer,
We wanted to thank you very much for your kind review. We proceeded to add, as suggested, a conclusive paragraph in the discussion that highlights the current limitations of ML use in the clinical practice (please see the attachment).
Kind Regards,
F Bonomi

Reviewer 2 Report
This is a well written review of machine learning implementation in systemic sclerosis.
In the introduction, the authors highlighted the clinical problem with the classification of systemic sclerosis and the problem of risk stratification.
The main part presents and characterizes the results of the literature review.
The discussion could be a bit more detailed and include application barriers and potential future applications.
The title should contain information about the review nature of the work, and at the end of the discussion there should be information about the limitations of the work - in this case, subjective assessment without applying the principles of systematic review.
Author Response
Dear Reviewer,
Thank you very much for your kind review of our paper. We proceeded, as suggested, to amplify the discussion part with a paragraph regarding the current limitations of ML use in clinical practice.
We also modified the title of the paper explaining that the current review is a narrative review and added a sentence regarding the limitation of the study.
Kind regards,
F Bonomi
Reviewer 3 Report
Dear authors,
thank you for allowing me to review your manuscript.
Following are some suggestions for improving your work:
In the introduction section, a connection between SS and Machine Learning would be useful. Maybe a zipper sentence between 57 and 58 lines (page 2)?
The Materials and Methods section is quite poor in the description.
I suggest better describing the study design, avoiding explaining what you did not do, and instead expanding on what you did and which author guided you in choosing the study design.
Thus, perhaps it would be helpful to explain who read the articles; how the data were handled and analyzed; whether and how an agreement was reached among the authors.
Research strings could be enriched with the mesh term "machine intelligence," even though I think it would not change the final outcome.
In Table 1, I would recommend removing the "reference" column as it is unnecessary
Finally, it might also be useful to highlight possible barriers or open issues revealed by the different studies concerning the use of ML-equipped technologies. This may allow you an opportunity to provide a wider perspective product (also in terms of future citations).
Best regards,
MEDC
Author Response
Dear Reviewer,
Thank you very much for your kind review. We proceeded to add, as suggested, a linking sentence in the introduction to create a connection between SSc and ML.
Regarding the Materials and Methods comment, you will find that we did not change the content because it is a non-systemic review. Therefore, since it lacked the systemic workup, it was difficult to explain more the methods. No datas was handled nor analyzed by us. We proceeded however to search also the mesh term "machine intelligence" and found a recent paper that was published at the end of may, and proceeded than to add it to the study.
Finally, we proceeded to remove the reference column from the table and to add the limitations of ML use in SSc in clinical practice at the end of the discussion.
Kind regards,
F Bonomi